# Predicting willingness to be vaccinated for Covid-19: Evidence from New Zealand

**Geoff Kaine** [ID] [1]☯*, **Vic Wright** [ID] [2]☯, **Suzie Greenhalgh** [3]☯

**1** Manaaki Whenua – Landcare Research, Hamilton, New Zealand, **2** University of New England, Armidale, New South Wales, Australia, **3** Manaaki Whenua – Landcare Research, Auckland, New Zealand

☯ These authors contributed equally to this work.
* kaineg@landcareresearch.co.nz

**Data Availability Statement:** All relevant data are within the manuscript and its Supporting information files.

## Abstract

Governments around the world are seeking to slow the spread of Covid-19 and reduce hospitalisations by encouraging mass vaccinations for Covid-19. The success of this policy depends on most of the population accepting the vaccine and then being vaccinated. Understanding and predicting the motivation of individuals to be vaccinated is, therefore, critical in assessing the likely effectiveness of a mass vaccination programme in slowing the spread of the virus. In this paper we draw on the $I_3$ Response Framework to understand and predict the willingness of New Zealanders to be vaccinated for Covid-19. The Framework differs from most studies predicting willingness to be vaccinated because it is based on the idea that the willingness to adopt a behaviour depends on both involvement (a measure of motivational strength) with the behaviour and attitudes towards the behaviour. We show that predictions of individuals' willingness to be vaccinated are improved using involvement and attitudes together, compared to attitudes alone. This result has important implications for the implementation of mass vaccination programmes for Covid-19.

## Introduction

The success of measures to slow or stop the spread of Covid-19, such as wearing masks, using tracer apps, and mass vaccination, depends on the commitment and capacity of individuals to accept them and change their behaviour accordingly. For example, failure to achieve high rates of vaccinations for Covid-19 may mean the recurrent imposition of lockdowns (which cause economic, social, and psychological damage) and continued investment of considerable resources in contact tracing, community testing, operating quarantine facilities, and the diversion of medical infrastructure and services to combat increased rates of infection and hospitalisation from Covid-19. Hence, understanding and predicting the extent to which individuals are motivated to participate in a mass vaccination programme is critical in assessing the likely effectiveness of the programme.

There is an extensive literature on people's willingness to be vaccinated, especially in regard to vaccine hesitancy [1–6], which shows that beliefs about, and attitudes towards, vaccines are fundamental to people's acceptance of vaccines and their willingness to be vaccinated.

**Funding:** This research was funded by the New Zealand Ministry for Business, Innovation and Employment (https://www.mbie.govt.nz/) through the Te Pūnaha Matatini – NZ COVID Modelling Programme (https://www.tepunahamatatini.ac.nz/). MWLR Client project number: UOAX1941. The funders had no role in study design, data collection and analysis, decision to publish, or preparation of the manuscript.

**Competing interests:** The authors have declared that no competing interests exist.

Consequently, this literature points to education and promotion as key strategies for changing awareness, beliefs, and attitudes to vaccine hesitancy and encouraging participation in vaccination programmes [3, 6–8]. The success of these strategies depends on how malleable people's attitudes are [9] and how attentive they are to education and promotion activities [10–12].

In this paper we draw on the $I_3$ Response Framework [6] to predict how strongly New Zealanders are motivated to be vaccinated for Covid-19, and to investigate differences in the strength and stability of New Zealanders' attitudes towards being vaccinated for Covid-19. We then draw out implications for how attentive New Zealanders might be to education and promotion activities that encourage their participation in a mass vaccination programme for Covid-19.

## Theory

Kaine et al. [13] proposed that theories about people's responses to policy measures, such as urgings to get vaccinated, implicitly presume that the measures are inherently important enough to people that they devote considerable cognitive effort to gathering information about the measure, processing that information, formulating attitudes towards the measure, and reaching a decision about whether to comply (in the broadest sense of the word) with the measure or not. Kaine et al. [13] suggested that these theories cannot be expected to predict behaviour accurately when this presumption is invalid, and when the subject (e.g. being vaccinated against Covid-19) is not perceived to be important enough (i.e. sufficiently relevant to people's personal goals) to trigger the effort required to form an attitude that has the power to influence their behaviour. Consequently, to predict how people may or may not respond to any given policy measure it is necessary to understand whether they are likely to invest effort in decision-making regarding that measure.

As explained in detail in Kaine et al. [13], the effort people will devote to decision-making about complying with a policy measure will depend on their *involvement* with the policy *issue* (in this case, the policy outcome of eliminating Covid-19 from New Zealand) and the *intervention* (the policy measure, such as being vaccinated for Covid-19), with the former being an important component of the context for the latter. These concepts underpin the $I_3$ Response Framework (*i*nvolvement, *i*ssue, and *i*ntervention, hence $I_3$) used in this analysis.

The behaviour changes to be analysed with the $I_3$ Framework occur in a public policy rather than a commercial context. This means the outcome(s) sought are typically declared, and government, or its agencies, intervene with measures designed to modify behaviour in pursuit of the outcome(s). Either compulsion or voluntary responses may be involved, but compliance is central to achieving outcomes. In what follows in this section we have drawn extensively on the discussion of the interpretation of Framework findings from Kaine et al. [13] to make it readily accessible for the reader.

### The $I_3$ framework

Kaine et al. [13] proposed that people's responses to policy measures can be inferred from their:

- involvement with the relevant policy outcome which, in this context, is involvement with preventing the spread of Covid-19 (not involvement with Covid-19 per se)

- involvement with, and attitude towards, the policy measure itself which, in this context, is getting vaccinated against Covid19.

Involvement with the policy measure signals the degree to which the measure itself is a source of motivation for the individual, irrespective of the policy issue [14, 15]. This allows for

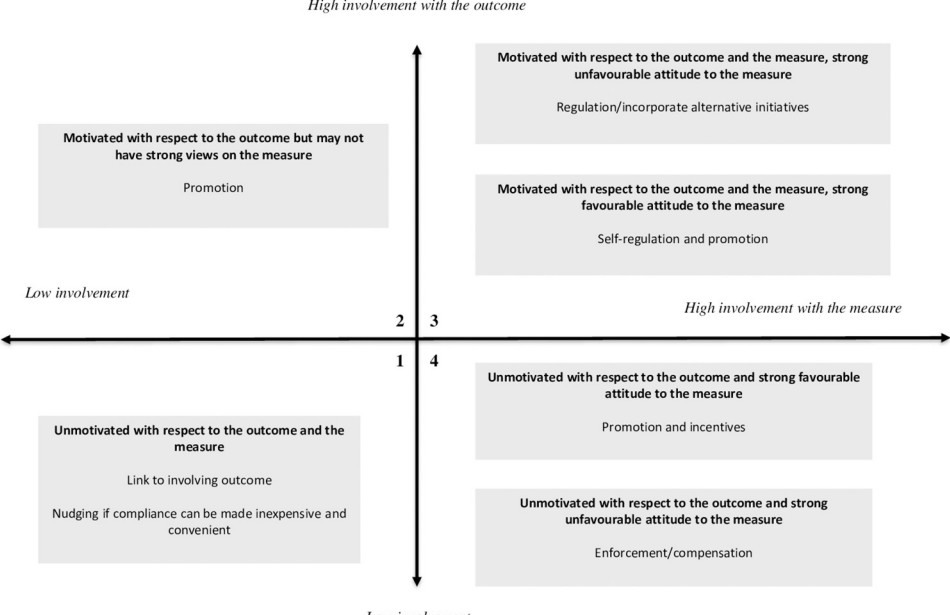

**Fig 1. Map and explanation of quadrants in the I$_3$ response framework.**

the possibility that individuals are motivated to act in response to a measure even though they do not perceive that the policy outcome the measure addresses is relevant to them. In such situations it may be that the wish to comply is motivated by involvement with some other outcome, such as conforming with social norms [16].

However, the perceived relevance of a policy outcome is relevant to an individual's cognitions about related measures. One would normally expect a positive correlation between involvement with the outcome and the measure while recognising the possibility of variation in involvement across various measures. The value of the Framework is that it enables a decomposition of overall involvement with a policy outcome and corresponding measures, as well as distinguishing between involvement with different measures and closer analysis of the role of beliefs held by individuals, as informational contexts for attitudes. This has implications for the relevance of the various components of vaccination hesitancy, which we refer to in our description of the Framework below.

The two dimensions of involvement with the policy outcome and involvement with the policy measure mean that the responses of people to a policy measure can be classified into four quadrants, as shown in Fig 1.

People in quadrant 1 exhibit low involvement with both the policy outcome and the policy measure. Consequently, these people are hypothesised to have little knowledge, or even awareness, of the policy outcome and are likely to have limited knowledge of the policy measure, and to have weak attitudes towards it, if any at all. These people may be either detached because they have other interests and concerns, or they are 'know-nothings': they do not particularly care about or have any interest in the outcome [17]. Chaffee and Roser [18, p. 376] describe this behaviour as being 'a direct response to situational constraints and not especially reflective of one's attitudes or knowledge.' Therefore, for these people, non-compliance with the measure is largely unintentional [19].

In the context we are considering here, the low involvement of people in quadrant 1 with the policy outcome of eliminating Covid-19 and getting vaccinated has important implications

for overcoming vaccine hesitancy. In recent models of vaccine hesitancy such as the 5C model, hesitancy is decomposed into five components: confidence, complacency, constraints, calculation, and collective responsibility [5]. According to Wismans et al. [6]:

- confidence relates to trust in the effectiveness and safety of vaccines, in the system that delivers them, and in the motivations of policymakers;

- complacency reflects the perceived risk and perceived level of threat of vaccine-preventable diseases;

- constraints are psychological and physical barriers, such as accessibility, health literacy, and affordability;

- calculation relates to individuals' engagement in extensive information searching; and

- collective responsibility reflects a willingness to protect others by getting vaccinated.

Confidence and complacency, and potentially constraints and collective responsibility, depend on awareness and beliefs. Given their lack of interest, people in quadrant 1 could be expected to be indifferent with respect to confidence in vaccines, complacent with respect to personal risk, sensitive to constraints on being vaccinated, unengaged with calculation, and indifferent to collective responsibility.

If the proportion of people in quadrant 1 is low enough to present little threat to achieving the aggregate policy outcome of preventing the spread of Covid-19, they can be ignored [13]. Otherwise, their compliance with the measure (getting vaccinated) may be encouraged by:

- linking the policy outcome to a subject they find more involving (e.g. concern for family, employment, recreation, international travel);

- reducing the effort required to be compliant (e.g. offering vaccinations in convenient locations such as shopping malls, churches, and sports venues, or offering free vaccinations);

- offering incentives (e.g. cash and other rewards);

- promoting awareness of, and building knowledge about, the policy outcome and the policy measure.

However, because people in this quadrant are uninterested, they are unlikely to pay attention to promotional and educational messaging, so the final strategy of promoting awareness is likely to be ineffective. Kim [14] suggests that an affect-evoking strategy (i.e. one that evokes an emotional response) should be the most effective means of attracting people's attention under these circumstances. This is likely to be achieved by focusing the affect-evoking strategy on the policy outcome, as greater involvement with the outcome is likely to engender greater involvement with individual measures.

People in quadrant 2 exhibit high involvement with the policy outcome but low involvement with the measure. Consequently, they would be aware of the outcome and invest time and energy in processing information, decision-making, and responding to the outcome [18, 20]. They may have limited knowledge of the policy measure and may have weak or ambivalent attitudes towards it. Any non-compliance with the measure is largely unintentional [13].

Regarding vaccine hesitancy [6], people in quadrant 2 could be expected to be unsure or indifferent with respect to confidence in vaccines, may be complacent with respect to personal risk because they are young and healthy and/or they feel the likelihood of exposure is low because the incidence of transmission in the community is low. They may be somewhat

sensitive to constraints on being vaccinated, modest on calculation, but have some sense of collective responsibility.

If people in quadrant 2 represent little risk in terms of achieving the policy outcome, they can be ignored. If their compliance is important to achieving the policy outcome, reducing the effort required for compliance [21] and promoting awareness of the policy measure may be worthwhile. This can be done by taking advantage of the intensity of their involvement with the policy outcome, particularly when this is accompanied by favourable attitudes towards the measure.

People in quadrant 3 exhibit high involvement with both the policy outcome and the measure. These people are likely to have extensive and detailed knowledge of the policy outcome. They are also likely to have extensive knowledge of the policy measure and strong attitudes towards it [20]. If their attitude towards the policy measure is favourable, they will comply with the measure and may even advocate for it [13]. Consequently, a strategy for promoting compliance among individuals in this quadrant with a favourable attitude might focus on self-regulation. Promotion and monitoring may also be worthwhile to ensure awareness and knowledge of obligations, ensure desirable behaviours are maintained, and identify at an early stage any changes in their attitude [13].

In terms of vaccine hesitancy [6], people in quadrant 3 could be expected to be polarised. If people in this quadrant had a favourable attitude towards vaccines then we would expect them to have high confidence in vaccines, be non-complacent with respect to personal risk, be insensitive to constraints on being vaccinated, be thoroughly engaged with calculation, and have a strong sense of collective responsibility. People in this quadrant with an unfavourable attitude towards vaccines could be expected to have low confidence in vaccines, be non-complacent with respect to personal risk, be insensitive to constraints on being vaccinated, be thoroughly engaged with calculation, and have a weak sense of collective responsibility in terms of willingness to protect others by getting vaccinated.

If people in quadrant 3 have an unfavourable attitude towards the policy measure, they may comply, but reluctantly [13]. Non-compliance with the measure will be intentional. Most likely they will prefer–and even advocate for–an alternative policy measure. Where practical, incorporating these changes may encourage compliance among these people [22]. Alternatively, offering incentives to reduce compliance costs may neutralise unfavourable reactions.

Another strategy for promoting compliance among people in this quadrant with an unfavourable attitude is to change their attitude towards the measure. This may be possible by reframing the benefits about the measure in terms of another subject that is more involving for them [13], thus provoking a recalculation of net costs and benefits. Alternatively, a promotional programme could be implemented with the outcome of persuading these people that they are mistaken, and that the behaviour required by the policy measure (being vaccinated) is superior to any alternatives. This strategy is likely to fail if people in this quadrant hold strongly favourable, or unfavourable, attitudes as they are likely to engage in motivated reasoning [11, 12]; i.e., filtering out information that challenges their beliefs and attitudes. Finally, compliance among these individuals might be increased by investing resources in enforcement, to increase the likelihood of detection and prosecution, and legislating severe penalties for non-compliance. However, authorities would need to carefully consider the imposition of blanket penalties for non-compliance because they run the risk of incidentally alienating people with low involvement.

Note that if the causes of non-compliance relate to unpredictable variations in the environment, or to unforeseeable technical problems, then enforcement and general deterrence may be ineffective. A more appropriate strategy in these circumstances may be to focus on the provision of technical assistance [23, 24].

People in quadrant 4 exhibit low involvement with the policy outcome but high involvement with the measure. People in this quadrant are likely to have limited knowledge of the policy outcome. They are likely to have detailed knowledge of the policy measure and have strong attitudes towards it [20]. If their attitude towards the measure is favourable, they will comply with the measure [13]. In these circumstances the government agency may play a monitoring role to check that the conditions promoting compliance do not change. A promotional strategy to support and reinforce compliance behaviour may also be worthwhile.

On the other hand, if the members of this quadrant have an unfavourable attitude towards the policy measure, they will only comply reluctantly, or may intentionally refuse to comply at all. These people will regard the measure as intrusive and as imposing unwarranted costs upon them. Most likely they will agitate against the policy measure [13] because they are not committed to the outcome. One strategy for promoting compliance among these individuals is to change their attitude towards the measure. This may be possible by reframing it in terms of another, more involving subject [13]. Offering incentives to offset compliance costs, or delaying or staging the introduction of policy measures, may neutralise unfavourable reactions [22]. Finally, compliance among these individuals might be increased by investing resources to increase the likelihood of detection and prosecution of non-compliance, and by introducing severe penalties. Again, as mentioned earlier, authorities would need to carefully consider the imposition of blanket penalties for non-compliance because they run the risk of incidentally alienating people with low involvement.

With respect to vaccine hesitancy [6], people in quadrant 4 could be expected to be polarised. People in this quadrant with a favourable attitude towards vaccines could be expected to have high confidence in vaccines, be variably complacent with respect to personal risk, be insensitive to constraints on being vaccinated, be thoroughly engaged with calculation, and be variable in their sense of collective responsibility (especially those who view vaccination purely in terms of personal protection). People in this quadrant with an unfavourable attitude towards vaccines could be expected to have low confidence in vaccines, be variably complacent with respect to personal risk, be insensitive to constraints on being vaccinated, be thoroughly engaged with calculation, and have a weak sense of collective responsibility in terms of willingness to protect others by getting vaccinated.

In summary, Kaine et al. [13] hypothesised that individual responses to policy measures will depend on the intensity and source of involvement of the individual with the measure and, where that involvement is sufficiently intense to form an attitude, on whether that attitude is favourable or unfavourable. In a specific applied setting, such as a policy to control the spread of Covid-19, the $I_3$ Framework enables the prediction of people's likely compliance with measures such as willingness to participate in a mass vaccination programme and, given the reasons for their involvement and their attitudes, the best ways to enhance that compliance. The $I_3$ Framework has been employed to understand and predict compliance behaviour in a variety of contexts in agriculture [19, 23, 25, 26], rural and urban predator control [27, 28], and community support for predator control [29, 30].

## Covid-19 in New Zealand

Covid-19 was first detected in New Zealand on 28 February 2020 [31]. Within 3 weeks the central government had closed New Zealand's international border to all except returning citizens and permanent residents. The government began pursuing a restrictive strategy [32] of eliminating Covid-19 and applied a range of control measures to stop the transmission of Covid-19 in New Zealand [33]. Elimination did not necessarily mean eradicating the virus permanently from New Zealand; rather, that central government was confident chains of transmission in

the community had been eliminated for at least 28 days, and any cases imported from overseas in the future could be effectively contained [33].

The central government commenced a mass vaccination programme for Covid-19 using the Pfizer vaccine, starting with border and managed isolation and quarantine workers, in February 2021 [34]. The programme was accompanied by an extensive, government-funded publicity campaign using traditional and social media. The survey in this study was completed during the first and second week of March 2021 before vaccinations were available to the general public.

## Materials and methods

Given the reasoning underpinning the $I_3$ Response Framework [13], our purpose in this paper was to investigate the extent to which the willingness of New Zealanders to be vaccinated for Covid-19 depended on their involvement (a measure of motivational strength) with the idea of eliminating Covid-19 and with the idea of being vaccinated, as well as their attitudes (which depend on their beliefs) towards these ideas. Consequently, a questionnaire seeking information from the public on their beliefs about, attitudes towards, and willingness to be vaccinated against, Covid-19 was designed based on the $I_3$ Response Framework [13]. The questionnaire is reproduced in S1 File.

Involvement with eliminating Covid-19 from New Zealand and involvement with getting vaccinated against Covid-19 were each measured using a condensed version of the Laurent and Kapferer [35] involvement scale developed by Kaine [36], with respondents rating two statements on each of the five components of involvement (functional, experiential, identity-based, consequence-based, and risk-based). Functional involvement arises from utilitarian and economic needs and experiential involvement arises from pleasure needs. Identity involvement concerns self-concept and impression management needs. The intensity of involvement is also be amplified by an individual's perception of the seriousness of making mistakes in relation to those needs, and the risks of making such mistakes [13].

Attitudes were measured using a simple, evaluative scale, while the strength of respondents' attitudes, which were expected to vary depending on the strength of their involvement, was measured using an ipsative scale based on Olsen [37]. A series of questions was formulated to elicit respondents' beliefs about Covid-19, eliminating Covid-19, and getting vaccinated for Covid-19.

The ordering of the statements within each of the involvement, attitude, and belief scales was randomised to avoid bias in responses. Respondents indicated their agreement with statements in all the involvement, attitude, and belief scales using a five-point rating, ranging from strongly disagree (1) to strongly agree (5). Involvement scores were computed as the simple arithmetic average of respondents' agreement ratings for the ten statements in the involvement scales. Attitude scores were computed as the simple arithmetic average of respondents' agreement ratings for the five statements in the attitude scales. Belief scores were simply respondents' agreement ratings on each belief statement. The internal consistency of the involvement and attitudinal scales was assessed using Cronbach's alpha [38].

Respondents were asked whether they were willing to get vaccinated against Covid-19 and to indicate their willingness using a five-point rating, ranging from 'definitely not' (1) to 'definitely' (5). Respondents who answered that they might, probably would or definitely would get vaccinated were also asked if they would get vaccinated as soon as possible and whether they would get vaccinated even if the vaccine only offered a few months protection.

Respondents were questioned about the policy outcome of eliminating Covid-19 from New Zealand. They were asked whether they felt some responsibility for eliminating Covid-19 from New Zealand, were prepared to change their normal behaviour or to make sacrifices to

eliminate Covid-19 from New Zealand and if it was important to work together to eliminate Covid-19 from New Zealand. Respondents indicated their agreement with these policy outcome statements using a five-point rating, ranging from 'strongly disagree' (1) to 'strongly agree' (5).

Information was sought on the demographic characteristics of respondents, including age, education, and ethnicity, and whether they were willing to be vaccinated for Covid-19. The ethnicity categories were Māori (the indigenous people of New Zealand), European New Zealander, Pacific Islander, Asian, and Other.

Participation in the survey was voluntary, respondents could leave the survey at any time, and all survey questions were optional and could be skipped. The research approach was reviewed and approved by social ethics process of Manaaki Whenua–Landcare Research (application no. 2021/27), which is based on the New Zealand Association of Social Science Research code of ethics.

The questionnaire, stratified by regional population, was distributed from 4 March to 15 March 2021 to a random sample of New Zealanders who were members of an online consumer panel. Panel members receive reward points (which are redeemable for products and services) for completing surveys. A total of 1,002 completed responses were obtained within the two-week period of which 53% were from women and 47% from men.

The age distribution aligned closely with the 2018 Census distribution, but Māori and Pacific Island residents were under-represented in the sample while 'other' ethnicities were over-represented. The household income distribution of the sample approximated the distribution from the 2018 Census. Residents with secondary or certificate qualifications were substantially under-represented in the sample, while residents with graduate and postgraduate qualifications were substantially over-represented (see S1 Appendix).

At the time the survey was conducted Auckland residents were under Alert Level 2, which meant they were expected to maintain social distancing when outside their homes and to wear masks in public places. They were also expected to keep track of their movements and to self-isolate and seek testing for Covid-19 if they felt unwell and experienced symptoms associated with Covid-19. The rest of the country was at Alert level 1 during this time. At Alert Level 1 social distancing and the wearing of face masks in public places is not expected. People are encouraged to keep track of their movements and to self-isolate and seek testing for Covid-19 if they feel unwell and experience symptoms associated with Covid-19.

Given these circumstances and given that the pandemic had been receiving widespread coverage by traditional and social media in New Zealand since February 2020, it seems reasonable to suppose that virtually all respondents were aware of Covid-19 at the time we conducted our survey and that most, if not all, were also aware of the government intention to institute a mass vaccination programme. While awareness of the existence of Covid-19 is a prerequisite for involvement with it, awareness does not necessarily entail involvement. Widespread awareness of Covid-19 simply creates the potential for widespread involvement. The extent to which that potential is realised depends on respondents' beliefs about how Covid-19 could affect the achievement of their functional, experiential, and self-identity needs.

The data were analysed in three stages. First, the expected associations between $I_3$ quadrant membership and willingness to be vaccinated were investigated. Respondents were allocated to quadrants depending on whether their involvement scores were less than, or more than, the mid-point of the scale (i.e. 3.00) and differences investigated in respondents' willingness to be vaccinated were investigated across the quadrants. We also explored differences across the quadrants in variables relating to the 5C model of vaccine hesitancy. Statistically significant differences among quadrants were identified using chi-square and analysis-of-variance tests as appropriate.

The purpose of the second stage of the analysis was to conveniently summarise respondents' beliefs about Covid-19, eliminating Covid-19 and getting vaccinated for Covid-19 for inclusion in the third stage of the analysis. Respondents were classified into three sets of segments using cluster analysis. The first set of segments was based on respondents' beliefs about Covid-19, the second set was based on their beliefs about eliminating Covid-19 and the third set was based on their beliefs about Covid-19 vaccines. Respondents were clustered into belief segments based on their agreement ratings with the set of relevant belief statements using Ward's method, with squared Euclidean distance as the measure of dissimilarity [39]. The number of segments was chosen based on the relative change in fusion coefficients, ease of interpreting the segments, and a desire to keep the number of segments as small as possible [39]. Statistically significant differences in beliefs among segments were identified using Tukey's HSD test [40].

In the third stage, using regression analysis, we quantified the effect of beliefs, attitudes, and involvement on respondents' support for the policy outcome of eliminating Covid-19 from New Zealand and on their propensity to be vaccinated for Covid-19. For this stage dummy variables were created representing respondents' membership of the belief segments identified in stage two with respect to Covid-19, eliminating Covid-19 and getting vaccinated for Covid-19. For each set, the relevant 'sceptics' segment was treated as the benchmark. Ordinary Least Squares and Binary logistic regressions were used in this stage of the analysis.

Statistical analyses were conducted using the 'cluster' and 'regression' commands in SPSS [41].

## Results

### Stage 1: $I_3$ membership and willingness to be vaccinated

As shown in Table 1, all of the involvement and attitudinal scales exhibited satisfactory reliabilities [38]. For convenience degrees of involvement were categorised as low (score less than 2), mild (score greater than 2 but less than 3), moderate (score greater than 3 but less than 4) and high (score greater than 4). Most respondents had moderate-to-high involvement with eliminating Covid-19 from New Zealand and with getting vaccinated and so were placed in quadrant 3 (Table 2). However, a substantial minority of respondents had low-to-mild involvement with getting vaccinated for Covid-19 and so were placed in quadrants 1 and 2.

A majority of respondents exhibited a strongly favourable attitude, as measured by the ipsative attitude scale, towards being vaccinated for Covid-19 (Table 3). These results suggest that only a small proportion of respondents would deliberately choose not to be vaccinated to protect themselves and help prevent the spread of Covid-19 in the community.

Consistent with involvement theory, a relatively high proportion of respondents in quadrant 1 and quadrant 2 were unsure whether getting vaccinated was the right thing to do. Consequently, a relatively low proportion of these respondents indicated they were likely to be willing to be vaccinated (Table 4), and those that were willing were less sure of being

**Table 1. Reliability of involvement and attitude scales.**

|  | Reliability |
|---|---|
| Involvement with eliminating Covid-19 | 0.847 |
| Involvement with being vaccinated for Covid-19 | 0.852 |
| Attitude towards being vaccinated for Covid-19 | 0.854 |

Notes: Values are Cronbach's alpha [38].

**Table 2. I$_3$ classification for being vaccinated for Covid-19.**

|  | Percentage of respondents |
|---|---|
| Quadrant 1 | 4.8 |
| Quadrant 2 | 7.4 |
| Quadrant 3 | 86.7 |
| Quadrant 4 | 1.1 |
| **Total** | 100.0 |

**Table 3. I$_3$ vaccine classification and attitude towards being vaccinated.**

| Attitude | Quadrant 1 | Quadrant 2 | Quadrant 3 | Quadrant 4 | Sample |
|---|---|---|---|---|---|
| Right thing to do | 12.5 | 21.6 | 74.0 | 54.5 | 67.0 |
| Doesn't matter to me | 8.3 | 9.5 | 5.3 | 9.1 | 5.8 |
| Not sure | 35.4 | 44.6 | 16.7 | 9.1 | 19.6 |
| Haven't given it much thought | 12.5 | 4.1 | 3.0 | 9.1 | 3.6 |
| Bad thing to do | 31.3 | 20.3 | 1.0 | 18.2 | 4.1 |

Notes: Values are percentages of respondents in each quadrant. Test for differences in percentages across quadrants ($\chi 2 = 259.1$, $P < 0.01$).

**Table 4. I$_3$ vaccine classification and willingness to be vaccinated.**

|  | Quadrant 1 | Quadrant 2 | Quadrant 3 | Quadrant 4 | Sample |
|---|---|---|---|---|---|
| Definitely | 12.5 | 9.5 | 56.7 | 36.4 | 50.9 |
| Probably | 6.3 | 16.2 | 20.0 | 18.2 | 19.1 |
| Maybe | 16.7 | 24.3 | 16.7 | 18.2 | 17.3 |
| Probably not | 16.7 | 23.0 | 4.9 | 9.1 | 6.9 |
| Definitely not | 47.9 | 27.0 | 1.6 | 18.2 | 5.9 |

Notes: Values are percentages of respondents in each quadrant or total sample. Test for differences in percentages across quadrants ($\chi^2 = 321.6$, $P < 0.01$).

vaccinated immediately once vaccines were available (Table 5) or to be vaccinated if vaccines only offered a few months' protection (Table 6). By contrast, and consistent with theory, a high proportion of the respondents in quadrant 3 were sure that being vaccinated was the right thing to do (Table 3), were willing to be vaccinated (Table 4), and as quickly as possible, once vaccines were available (Table 5). A high proportion of these respondents was willing to be vaccinated if vaccines only offered a few months' protection (Table 6). These results and their implications for government policy are discussed in detail by Kaine [42].

**Table 5. I$_3$ vaccine classification and willingness to be vaccinated as soon as possible.**

|  | Quadrant 1 | Quadrant 2 | Quadrant 3 | Quadrant 4 | Sample |
|---|---|---|---|---|---|
| Yes | 41.2 | 18.9 | 68.2 | 75.0 | 65.7 |
| Not sure | 29.4 | 29.7 | 21.6 | 25.0 | 22.1 |
| No | 29.4 | 51.4 | 10.2 | 0.0 | 12.2 |

Notes: Values are percentages of respondents in each quadrant (or the total sample) who indicated they definitely or probably would, or might, get vaccinated. Test for differences in percentages across quadrants ($\chi^2 = 70.3$, $P < 0.01$).

**Table 6. I₃ vaccine classification and willingness to be vaccinated even if it only offers a few months' protection.**

|          | Quadrant 1 | Quadrant 2 | Quadrant 3 | Quadrant 4 | Sample |
|----------|------------|------------|------------|------------|--------|
| Yes      | 33.3       | 27.8       | 61.2       | 50.0       | 59.8   |
| Not sure | 25.0       | 33.3       | 26.3       | 37.5       | 26.6   |
| No       | 41.7       | 38.9       | 12.5       | 12.5       | 13.6   |

Notes: Values are percentages of respondents in each quadrant (or the total sample) who indicated they definitely or probably would, or might, get vaccinated and excludes those who indicated they would not get vaccinated as a soon as possible. Test for differences in percentages across quadrants ($\chi^2 = 21.7$, $P < 0.01$).

The predominantly moderate-to-high involvement with, and favourable attitudes of most respondents towards, being vaccinated to help eliminate Covid-19 from New Zealand is consistent with the high level of community endorsement of the New Zealand Government's approach to managing Covid-19 [43] and the cumulative proportion of the population that has been vaccinated or is booked to be vaccinated [34].

## I₃ membership and the 5C model

Although this study was not designed to investigate relationships between the I₃ Framework and models of vaccine hesitancy such as the 5C model, and bearing in mind that we were not using the standard scales that have been developed for that model [5], we were able to examine some of the suggestions made earlier about these relationships. We found differences in beliefs broadly in line with our expectations about differences in confidence, complacency, constraints, and collective responsibility across the quadrants (Tables 7 and 8). We also found differences across the quadrants in media use that aligned with our expectations about differences in calculation.

For example, compared to respondents in quadrant 3, respondents in quadrant 1 appeared:

- less confident about the safety of Covid-19 vaccines (confidence);

**Table 7. I₃ vaccine classification, confidence, complacency, and constraints.**

|  | Quadrant 1 | Quadrant 2 | Quadrant 3 | Quadrant 4 |
|---|---|---|---|---|
| Confidence:<br>It isn't worth getting vaccinated against Covid-19 yet as there are too many unknowns about the vaccines | 4.00 | 3.54 | 2.61 [a] | 3.09 |
| Children shouldn't be vaccinated against Covid-19 | 3.52 | 3.14 | 2.45 [a] | 2.64 |
| I think we should wait and see if the Covid-19 vaccination works overseas before trying it here | 3.50 | 3.61 | 2.83 [a, b] | 3.36 |
| Getting vaccinated against Covid-19 is unsafe because of the potential side-effects | 3.48 | 3.39 | 2.51 [a, b] | 2.82 |
| Complacency:<br>I think Covid-19 is a hoax | 2.38 | 1.95 | 1.57 [a, b] | 2.18 |
| Fears about Covid-19 are exaggerated | 3.83 | 2.73 [a] | 2.15 [a, b] | 3.36 [c] |
| Covid-19 is no worse than the seasonal flu | 3.10 | 2.47 [a] | 1.99 [a, b] | 2.18 |
| Constraints:<br>Getting vaccinated against Covid-19 is just not practical | 2.98 | 2.27 | 2.14 [a, b] | 2.73 |
| Getting vaccinated against Covid-19 isn't worthwhile if you are only protected for a few months | 3.77 | 3.43 | 2.71 [a, b] | 3.18 |
| Getting vaccinated against Covid-19 is a waste of time and effort | 3.10 | 2.81 | 1.99 [a, b] | 2.64 |

Notes: Agreement with a statement was rated on a five-point scale from strongly disagree (1) to strongly agree (5).

Differences in mean agreement ratings between segments tested using Tukey's HSD [40], ($P < 0.01$).

[a] Mean agreement rating significantly different from mean for quadrant 1.

[b] Mean agreement rating significantly different from mean for quadrant 2.

[c] Mean agreement rating significantly different from mean for quadrant 3.

**Table 8. I₃ vaccine classification, calculation, and collective responsibility.**

| | Quadrant 1 | Quadrant 2 | Quadrant 3 | Quadrant 4 |
|---|---|---|---|---|
| Calculation:<br>Respondents who did not watch, read, or listen for news about Covid-19 on traditional media (%) [a] | 33.3 | 20.3 | 10.2 | 27.3 |
| Respondents who did not watch, read, or listen for news about Covid-19 on social media (%) [b] | 70.8 | 51.4 | 50.3 | 54.5 |
| Respondents who did not chat about Covid-19 with family, friends, or co-workers in the past week (%) [c] | 45.8 | 31.1 | 22.1 | 45.5 |
| Number of modes of traditional media (e.g., television, radio, magazines) viewed | 0.88 | 1.27 | 1.53 [d] | 1.09 |
| Number of social media sources (e.g., Facebook, Instagram) viewed | 0.37 | 0.72 | 0.85 [d] | 0.64 |
| Collective responsibility:<br>We need to eliminate Covid-19 from New Zealand to save lives | 2.54 | 3.89 | 4.22 [d, e] | 2.55 [e, f] |
| We should just live with Covid-19 until we have a vaccine | 3.42 | 2.74 [d] | 2.53 [d] | 3.36 |
| It would be better to let Covid-19 spread and build herd immunity | 3.54 | 2.43 [d] | 1.99 [d, e] | 3.09 [f] |
| You should only have to get vaccinated if you are old or have a health problem | 2.79 | 2.53 | 2.18 [d, e] | 2.45 |

Notes: Agreement with a statement was rated on a five-point scale from strongly disagree (1) to strongly agree (5).

Differences in mean agreement ratings between segments tested using Tukey's HSD [40], ($P < 0.01$).

[a] Test for differences in percentages across quadrants ($\chi^2 = 29.8$, $P < 0.01$).

[b] Test for differences in percentages across quadrants ($\chi^2 = 7.7$, $P < 0.05$).

[c] Test for differences in percentages across quadrants ($\chi^2 = 19.0$, $P < 0.01$).

[d] Mean agreement rating significantly different from mean for quadrant 1.

[e] Mean agreement rating significantly different from mean for quadrant 2.

[f] Mean agreement rating significantly different from mean for quadrant 3.

- to regard Covid-19 as less threatening (complacency);

- to believe being vaccinated was impractical and not worthwhile (constraints);

- to consult fewer traditional and social media (calculation); and

- less sure that healthier and younger people needed to be vaccinated (collective responsibility)

## Stage 2: Belief segments

Respondents' beliefs were investigated because they can provide insights to guide the design of policies that, by modifying the beliefs and attitudes that underlie compliance, seek to influence compliance. Using respondent agreement ratings for the relevant set of belief statements, respondents were first classified into a set of segments with respect to their beliefs about the nature of Covid-19. Next, they were classified into a second set of segments based on their beliefs about eliminating Covid-19. Lastly, they were classified into a third set of segments based on their beliefs about the advantages and disadvantages of Covid-19 vaccines. The segments, and their belief characteristics, are summarised below.

Beliefs about Covid-19, eliminating Covid-19, and getting vaccinated for Covid-19 were associated, to some extent, with demographic characteristics such as age, education, income, and ethnicity (see [42] for details).

## Belief segments for Covid-19

Respondents were clustered into five belief segments with respect to Covid-19 (Table B1 in S2 Appendix). Note that the survey was conducted before the widespread emergence of the delta

variant of Covid-19. Most respondents had beliefs that align with accepted scientific facts. These respondents were classified as 'Covid-19 convinced' (37%) and 'Covid-19 moderates' (40%), the difference between these two segments being the intensity of their beliefs. The 'Covid-19 asymptomatics' (9%) had beliefs that mostly align with accepted scientific facts, but these respondents either disagreed that Covid-19 was spread by people coughing and sneezing or by contact with surfaces touched by infected people. This may reflect an awareness that Covid-19 can be transmitted by people with Covid-19 who are asymptomatic, and an awareness that the likelihood of infection by contact with contaminated surfaces is low. A fourth segment, the 'Covid-19 ambivalents' (12%), consisted of respondents who were unsure about what to believe about Covid-19. A small segment of respondents, the 'Covid-19 sceptics' (6%), believed that Covid-19 is a hoax, is no worse than the seasonal flu, and that fears about Covid-19 are exaggerated.

## Belief segments for eliminating Covid-19

Respondents were clustered into four belief segments with respect to eliminating Covid-19 (Table B2 in S2 Appendix). Most respondents have beliefs that align with seeking to eliminate Covid-19 from New Zealand. These respondents were classified as 'elimination enthusiasts' (26%) and 'elimination moderates' (18%), the difference between these two segments being the intensity of their beliefs. Another segment of respondents, the 'vaccination hopefuls' (34%), agreed with trying to eliminate Covid-19 but were less sure that Covid-19 could be kept out of New Zealand indefinitely. They believed we must live with Covid-19 until a vaccine is available. We did not define what the characteristics of a vaccine, once available, would be. A fourth segment, the 'elimination sceptics' (22%), consisted of respondents who believe we cannot eliminate Covid-19 indefinitely and we should try to build herd immunity.

## Belief segments for Covid-19 vaccination

Respondents were clustered into five belief segments with respect to being vaccinated against Covid-19 (Table B3 in S2 Appendix). Approximately half the sample was classified as 'vaccine enthusiasts' (24%) or 'vaccine moderates' (33%), the difference between these two segments being the intensity of their beliefs. These respondents believed that you will recover faster from Covid-19, and have weaker symptoms, if you are vaccinated. They were unsure whether being vaccinated stops you catching or spreading Covid-19 or gives you lifelong protection. They believed everyone should be vaccinated, and it should be compulsory and free. They did not believe Covid-19 vaccines are unsafe or that vaccinations should wait until experience overseas demonstrates that they work.

   A third segment of respondents, the 'vaccination cautious' (7%), fundamentally favoured being vaccinated but were concerned about side-effects. These respondents believed vaccination offers lifelong protection from contracting Covid-19, weakens symptoms, aids recovery, and prevents transmission. However, it appears that they believed vaccination should be limited at present to those who are at risk (the elderly and people with health problems), that children should not be vaccinated, and that vaccination should be free and compulsory once vaccines have been shown to be safe. This is because they believed that, at present, Covid-19 vaccines are unsafe, not enough is known about them, and vaccinations should wait until experience overseas demonstrates that they work.

   Another segment of respondents, the 'vaccination ambivalent' (28%), were fundamentally unsure whether vaccination offers protection from contracting Covid-19, weakens symptoms or aids recovery. They were unsure who should be vaccinated and whether vaccination should be compulsory, though they believed it should be free. They were unsure if Covid-19 vaccines

are safe or whether vaccinations should wait until experience overseas demonstrates they work.

The fifth segment consisted of the 'vaccination sceptics' (8%), who were not convinced vaccinations are beneficial. These respondents did not believe you would recover faster from Covid-19, or have weaker symptoms, if you were vaccinated. They did not believe being vaccinated stops you catching or spreading Covid-19 or gives you lifelong protection. Consequently, they thought vaccinations are a waste of time and effort and should not be compulsory. They believed that children should not be vaccinated, which is consistent with believing that Covid-19 vaccines are unsafe and too little is known about them. They believed vaccinations should wait until experience overseas demonstrates they work.

## Vaccine belief segments and being vaccinated

The respondents in the 'vaccine enthusiasts' and 'vaccine moderates' segments had a strongly favourable attitude towards being vaccinated (Table B4 in S2 Appendix). They were willing to be vaccinated, and as quickly as possible once a vaccine is available (Tables B5 and B6 in S2 Appendix). They were also willing to be vaccinated even if vaccination only offers protection for a few months (Table B7 in S2 Appendix).

When compared to the 'vaccination enthusiasts' and the 'vaccination moderates', a lower proportion of respondents in the 'vaccine cautious' segment had a strongly favourable attitude towards being vaccinated. Those in this segment who did think that being vaccinated is the right thing to do were likely to be vaccinated as quickly as possible once a vaccine (presumably one that is proven to be safe) is available. They were willing to be vaccinated even if vaccination only offers protection for a few months.

Most respondents in the 'vaccine ambivalent' segment are unsure about their attitude towards being vaccinated. As a result, they were unsure about being vaccinated, and were less likely to be vaccinated as quickly as possible once a vaccine is available. They were unwilling to be vaccinated if vaccination only offers protection for a few months.

Lastly, the respondents in the 'vaccine sceptics' segment had a strongly unfavourable attitude towards being vaccinated. Most were unwilling to be vaccinated, and those in this segment who would consider it would delay being vaccinated once a vaccine was available. They were generally unwilling to be vaccinated if the vaccine only offered protection for a few months.

A relatively high proportion of respondents in the 'vaccine cautious' and 'vaccine sceptics' segments reported having had a bad experience with vaccinations in the past and knowing someone else who had had a bad experience with vaccinations (Table B8 and B9 in S2 Appendix).

## Stage 3: Involvement, beliefs, attitudes, and willingness to be vaccinated

The purpose of this analysis was to quantify the effect of beliefs, attitudes and involvement on respondents' propensity to be vaccinated for Covid-19. In other words, we wanted to estimate, separately, the influence of involvement (as a measure of the strength of individuals' motivation) and the influence of beliefs and attitudes on willingness to be vaccinated in the context of Covid-19 in New Zealand.

Following Kaine et al. [13] we hypothesised that:

• respondents' attitudes towards eliminating Covid-19 are a function of beliefs about Covid-19 (as these condition the merit of having a policy response to Covid-19), beliefs about the worthiness of eliminating Covid-19 as a policy outcome, and beliefs about Covid-19 vaccinations as a long-term response to Covid-19;

- respondents' feelings of responsibility, willingness to change their normal behaviour, work with others and make sacrifices to eliminate Covid-19 are a function of their involvement with, and attitude towards, eliminating Covid-19, and their beliefs about Covid-19;

- respondents' attitudes towards being vaccinated for Covid-19 are a function of beliefs about Covid-19 and beliefs about Covid-19 vaccinations;

- the intensity of respondents' attitudes towards being vaccinated is a function of involvement with being vaccinated and beliefs about Covid-19;

- respondents' involvement with being vaccinated is a function of their involvement with eliminating Covid-19 and their beliefs about Covid-19;

- the willingness of respondents to act in support of eliminating Covid-19 from New Zealand is a function of involvement with, and attitude towards, being vaccinated (and beliefs about Covid-19);

- the willingness of respondents to be vaccinated for Covid-19 is a function of involvement with, and attitude towards, being vaccinated (and beliefs about Covid-19).

Respondents' propensity to be vaccinated was obtained by asking them, 'Once a vaccine for Covid-19 is available, will you get vaccinated?' Respondents answered the question using a five-point scale ranging from 'definitely' to 'definitely not'. They were also asked 'Once a vaccine is available would you get vaccinated as soon as you could?' and 'Would you get vaccinated if the vaccine only offered protection for a few months?' Both questions were answered as 'yes', 'no' or 'not sure', which were coded as binary variables (1 = yes, 2 = no or not sure) for the regression analysis.

Attitude toward eliminating Covid-19 as a strategy was measured as agreement with the statement 'Eliminating Covid-19 from New Zealand is the right thing to do' using a five-point scale from 'strongly disagree' (1) to 'strongly agree' (5). Feelings of responsibility, willingness to change normal behaviour, willingness to work with others, and willingness to make sacrifices to eliminate Covid-19 were measured as agreement with the relevant statement (see Table 11) using a five-point scale from 'strongly disagree' (1) to 'strongly agree' (5).

Attitude towards being vaccinated was measured by respondents' average score on the four-item attitudinal scale (see S1 File). Respondents' agreement with the statements in the attitudinal scale was rated using a five-point scale from 'strongly disagree' (1) to 'strongly agree' (5). The intensity or strength of attitudes was measured as the absolute value of respondents' attitude score after subtracting three (3), as this score signified 'unsure' in the rating scale for attitudes.

Dummy variables were created representing respondents' membership of belief segments with respect to Covid-19, eliminating Covid-19, and getting vaccinated. In each instance, the relevant 'sceptics' segment was treated as the benchmark.

The explanatory power of the regressions, and the resulting parameter estimates, are reported in Tables 9–12. The attitudinal regressions were statistically significant and, for cross-sectional data, a substantial proportion of the variance in the attitudes of respondents was explained by their beliefs. In all instances the signs on the estimated parameters were consistent with expectations, with attitudes becoming more and more unfavourable as respondents' beliefs shifted towards scepticism, as has been found in other studies [6, 8, 44, 45].

Respondents' attitude towards eliminating Covid-19 was strongly influenced by beliefs about the effectiveness of an elimination strategy and confidence in vaccination (Table 9). Respondents with ambivalent beliefs about vaccination (the 'vaccine cautious') were even more favourably disposed towards elimination than respondents who were more enthusiastic

**Table 9. Estimates of standardised parameters for attitudes towards eliminating Covid-19 and Covid-19 vaccination.**

| Variable | Attitude towards eliminating Covid-19 | Attitude towards getting vaccinated |
|---|---|---|
| | (n = 1002) | (n = 1002) |
| Covid-19 confident | 0.118 | 0.229 |
| | ($P = 0.138$) | ($P < 0.001$) |
| Covid-19 moderate | −0.009 | 0.186 |
| | ($P = 0.108$) | ($P < 0.001$) |
| Covid-19 ambivalent | −0.196 | 0.018 |
| | ($P < 0.001$) | ($P = 0.610$) |
| Covid-19 asymptomatic | −0.058 | 0.090 |
| | ($P = 0.150$) | ($P < 0.001$) |
| Elimination enthusiast | 0.413 | |
| | ($P < 0.001$) | |
| Elimination moderate | 0.136 | |
| | ($P < 0.001$) | |
| Elimination hopeful | 0.310 | |
| | ($P < 0.001$) | |
| Vaccination enthusiast | 0.247 | 1.107 |
| | ($P < 0.001$) | ($P < 0.001$) |
| Vaccination moderate | 0.205 | 0.996 |
| | ($P < 0.001$) | ($P < 0.001$) |
| Vaccination cautious | 0.251 | 0.400 |
| | ($P < 0.001$) | ($P < 0.001$) |
| Vaccination ambivalent | 0.158 | 0.510 |
| | ($P = 0.001$) | ($P < 0.001$) |
| Adjusted $R^2$ | 0.31 | 0.68 |
| F-test significance | <0.001 | <0.001 |

**Table 10. Estimates of standardised parameters for strength of attitudes towards eliminating Covid-19 and Covid-19 vaccination.**

| Variable | Strength of attitude towards eliminating Covid-19 | Strength of attitude towards getting vaccinated |
|---|---|---|
| | (n = 1002) | (n = 1002) |
| Covid-19 confident | 0.286 | 0.673 |
| | ($P < 0.001$) | ($P < 0.001$) |
| Covid-19 moderate | 0.092 | 0.434 |
| | ($P = 0.106$) | ($P < 0.001$) |
| Covid-19 ambivalent | 0.009 | 0.191 |
| | ($P = 0.846$) | ($P < 0.001$) |
| Covid-19 asymptomatic | 0.050 | 0.237 |
| | ($P = 0.150$) | ($P < 0.001$) |
| Elimination involvement | 0.426 | |
| | ($P < 0.001$) | |
| Vaccination involvement | | 0.250 |
| | | ($P < 0.001$) |
| Adjusted $R^2$ | 0.29 | 0.21 |
| F-test significance | <0.001 | <0.001 |

**Table 11. Estimates of standardised parameters for sense of responsibility, willingness to change behaviour, willingness to make sacrifices, and willingness to work with others to eliminate Covid-19.**

| Variable | I feel some responsibility for eliminating Covid-19 from New Zealand | I am prepared to change my normal behaviour to eliminate Covid-19 from New Zealand | I am prepared to make sacrifices to eliminate Covid-19 from New Zealand | It is important to work together to eliminate Covid-19 from New Zealand |
|---|---|---|---|---|
| Covid-19 confident | 0.050 | 0.234 | 0.084 | 0.351 |
| | ($P = 0.330$) | ($P < 0.001$) | ($P < 0.001$) | ($P < 0.001$) |
| Covid-19 moderate | −0.029 | 0.142 | 0.003 | 0.245 |
| | ($P = 0.579$) | ($P = 0.002$) | ($P = 0.947$) | ($P < 0.001$) |
| Covid-19 ambivalent | −0.100 | -0.003 | -0.090 | 0.069 |
| | ($P = 0.014$) | ($P = 0.924$) | ($P = 0.013$) | ($P = 0.050$) |
| Covid-19 asymptomatic | −0.042 | 0.058 | 0.039 | 0.080 |
| | ($P = 0.176$) | ($P = 0.043$) | ($P = 0.161$) | ($P = 0.013$) |
| Elimination involvement | 0.366 | 0.367 | 0.379 | 0.282 |
| | ($P < 0.001$) | ($P < 0.001$) | ($P < 0.001$) | ($P = 0.003$) |
| Elimination attitude | 0.282 | 0.341 | 0.353 | 0.437 |
| | ($P < 0.001$) | ($P < 0.001$) | ($P < 0.001$) | ($P < 0.001$) |
| Adjusted $R^2$ | 0.43 | 0.53 | 0.54 | 0.57 |
| F-test significance | <0.001 | <0.001 | <0.001 | <0.001 |

about vaccination, perhaps because elimination might reduce the necessity for vaccination. Beliefs about Covid-19 did not appear to strongly influence attitudes towards elimination over and above beliefs about elimination.

Respondents' attitude towards being vaccinated for Covid-19 was strongly influenced by beliefs about Covid-19 and beliefs about vaccination (Table 9). Respondents expressed increasingly unfavourable attitudes towards being vaccinated the more sceptical their beliefs about Covid-19 and the more ambivalent their beliefs about vaccination.

The regressions for intensity or strength of attitudes were statistically significant. A reasonable proportion of the variance in attitude strength of respondents was explained by their beliefs about Covid-19, their involvement with eliminating Covid-19, and their involvement with being vaccinated (Table 10). With respect to feelings of responsibility, willingness to change normal behaviour, willingness to work with others, and willingness to make sacrifices to eliminate Covid-19, involvement with, as well as attitude towards, elimination were significant in each regression, as were beliefs about Covid-19 (see Table 11).

As hypothesised, variations in respondents' involvement with being vaccinated depended on their involvement with eliminating Covid-19 and their beliefs about Covid-19 (Table 12).

**Table 12. Estimates of standardised parameters for involvement with Covid-19 vaccination.**

| Variable | Involvement with getting vaccinated (n = 1002) |
|---|---|
| Covid-19 confident | −0.358 ($P < 0.001$) |
| Covid-19 moderate | −0.359 ($P < 0.001$) |
| Covid-19 ambivalent | −0.265 ($P < 0.001$) |
| Covid-19 asymptomatic | −0.152 ($P < 0.001$) |
| Elimination involvement | 0.647 ($P < 0.001$) |
| Adjusted $R^2$ | 0.47 |
| F-test significance | <0.001 |

**Table 13. Parameter estimates for willingness to be vaccinated for Covid-19.**

| Variable | Willingness to be vaccinated [a] | Wanting to be vaccinated as quickly as possible [b] | Willing to be vaccinated even if benefits are temporary [b] |
|---|---|---|---|
| | (n = 1002) | (n = 874) [c] | (n = 767) [d] |
| Covid-19 confident | | 2.733 | 2.733 |
| | | (P<0.001) | (P<0.001) |
| Covid-19 moderate | | 2.322 | 2.680 |
| | | (p<0.001) | (p<0.001) |
| Covid-19 ambivalent | | 1.856 | 2.217 |
| | | (p<0.001) | (p<0.001) |
| Covid-19 asymptomatic | | 1.810 | 1.810 |
| | | (P = 0.010) | (P = 0.006) |
| Involvement with being vaccinated | 0.142 | 0.534 | 0.579 |
| | (p<0.001) | (p = 0.010) | (p = 0.003) |
| Attitude towards being vaccinated | 0.749 | 2.120 | 1.415 |
| | (p<0.001) | (p<0.001) | (p<0.001) |
| Intercept | | -16.298 | -14.762 |
| | | (p<0.001) | (p<0.001) |
| Adjusted $R^2$ | 0.70 | 0.44[e] | 0.29[e] |
| F-test significance | <0.001 | <0.01 | <0.01 |

Notes: Involvement scored as a rating from 1 to 5 (low to high involvement).

Attitude scored as a rating from 1 to 5 (unfavourable to favourable).

Willingness to be vaccinated scored as a rating from 1 to 5 (always to never).

Willing to be vaccinated as quickly as possible scored as a binary (1 = yes, 0 = no or not sure).

Willing to be vaccinated even if protection is temporary scored as a binary (1 = yes, 0 = no or not sure).

[a] Standardised parameter estimates reported for willingness to be vaccinated.

[b] Estimated as binary logistic regressions.

[c] Respondents who indicated they probably or definitely would not get vaccinated did not answer this question

[d] Respondents who indicated they probably or definitely would not get vaccinated, or who indicated they would delay getting vaccinated, did not answer this question.

[e] Nagelkerke R-Square.

While increasing scepticism about Covid-19 was associated with a more unfavourable attitude towards being vaccinated (Table 9), increasing scepticism about Covid-19 was associated with higher involvement with being vaccinated (Table 12). The greater involvement with vaccination associated with increasing scepticism reflects, perhaps, decreasing confidence in the benefits of vaccination, which translates into higher risk and consequence involvement, therefore higher overall involvement with Covid-19 vaccination as scepticism increases. Mean consequence and risk involvement were statistically significantly higher ($p < 0.01$) for respondents in the 'covid sceptics' segment compared to respondents in other segments.

The regressions concerning willingness to be vaccinated were also statistically significant (Table 13). They show that involvement with being vaccinated, in addition to attitude towards being vaccinated, explained a substantial proportion of the variance in respondents' willingness to be vaccinated. Respondents' desire to be vaccinated as quickly as possible once vaccines are available depended on their beliefs about Covid-19 as well as their involvement with, and attitude towards, being vaccinated. Similarly, willingness to be vaccinated, even though the effect might be temporary, was influenced by beliefs about Covid-19 as well as involvement with, and attitude towards, being vaccinated.

## Discussion

Our results clearly indicate that involvement with eliminating Covid-19, as well as beliefs about Covid-19, significantly influence feelings of responsibility, willingness to change normal behaviour, willingness to work with others, and willingness to make sacrifices to eliminate Covid-19. Our results show that involvement, a measure of motivational strength, contributes significantly to predictions of people's willingness to be vaccinated as well as their attitude towards being vaccinated. Also, our results show that involvement contributes significantly to predictions of the strength of people's attitude towards being vaccinated. The simple but important conclusion that follows from these results is that people may hold similar opinions or attitudes towards a protocol such as being vaccinated, but their propensity to do so may vary markedly depending on how involved they are with–how strongly they care about–preventing the spread of Covid-19 and getting vaccinated.

This conclusion has important implications for promoting community participation in mass vaccination programmes for Covid-19 to overcome the five components of vaccine hesitancy: confidence, complacency, constraints, calculation, and collective responsibility [6]. The first is related to the possibility that people may fail to comply with a measure even though they may have favourable attitudes towards the policy outcome, simply because they are not paying attention. In circumstances where involvement is low, compliance (or non-compliance) is not a matter of deliberate choice. This means that people who have low involvement with eliminating Covid-19 and with being vaccinated (quadrant 1) may fail to be vaccinated simply because it is inconvenient, not because they may have an unfavourable attitude towards being vaccinated. In terms of the 5C model, they are constrained.

Maximising vaccinations among people with low involvement requires ensuring that getting vaccinated requires as little effort and thought as is practical, and the experience is as stress-free as possible [46]. This means offering vaccinations at as many sites as possible, and at venues that are routinely visited such as shopping malls, churches, fast food outlets, workplaces, and sporting facilities. The aim is to minimise time spent travelling and queuing, and to create opportunities to be vaccinated on impulse. People with low involvement are likely to respond favourably to relatively small financial incentives if the process of being vaccinated is convenient and quick.

Relatedly, authorities should carefully consider the imposition of blanket penalties for non-compliance because they run the risk of incidentally alienating people with low involvement. For example, mandating workplace vaccinations for Covid-19 or the wearing of full personal protection equipment (PPE) by the unvaccinated may generate resentment, and possibly resistance, among those with low involvement (quadrant 1) irrespective of their attitude, albeit weak, towards Covid-19. Ensuring vaccinations can be obtained quickly and conveniently would be essential to alleviating feelings of resentment among those with low involvement.

In circumstances where involvement is high, compliance (or non-compliance) is most likely to be a deliberate choice, depending on one's attitude. With respect to being vaccinated for Covid-19, if attitudes towards vaccination are strongly unfavourable, then severe penalties or substantial inducements may be required to secure compliance. For example, in the workplace this may mean compelling staff either to be vaccinated or to wear full PPE and submit to a rigorous testing regime; and similarly, requiring international travellers to self-isolate in quarantine for a lengthy period (at their cost) if they are unvaccinated.

In relation to people with an unfavourable attitude towards being vaccinated, it is important to distinguish people who have high involvement with being vaccinated and with eliminating Covid-19 (quadrant 3) from people who have high involvement with being vaccinated but low involvement with eliminating Covid-19 (quadrant 4). In terms of the 5C model, the former

(quadrant 3) may be encouraged to seek vaccination using promotional messages aimed at reducing their complacency, building their confidence in Covid-19 vaccines, and appealing to their sense of collective responsibility. The latter (quadrant 4) are unlikely to respond favourably to such promotional messages because they have little interest in preventing the spread of Covid-19. Consequently, they are unlikely to respond to an appeal to their sense of collective responsibility.

Theoretically, people who have high involvement with being vaccinated but low involvement with eliminating Covid-19, and an unfavourable attitude towards being vaccinated (quadrant 4), are likely to be most confrontational in their opposition to being vaccinated. As they have little, if any, interest in the policy outcome of eliminating Covid-19, there is no justification, in their view, for forcing them to be vaccinated. Clearly, differences in the level of involvement people have with the outcome of eliminating Covid-19 and with being vaccinated to prevent the spread of Covid-19 create an additional complication for authorities.

Another implication concerns the possibility that people who have low involvement with the policy outcome and the policy measures may miss important promotional messages simply because they are not paying attention. In circumstances where involvement with a subject is low, exposure and sensitivity to promotional messages about the subject is low. Messages are not necessarily deliberately ignored; they simply fail to catch the attention of those with low involvement (they are just not noticed). This means, in circumstances where involvement is low, promotional efforts intended to increase the rate of vaccinations are unlikely to succeed because these efforts will, largely, be ignored. The lack of interest in eliminating Covid-19 and in being vaccinated (quadrant 1) means those with low-to-mild involvement will, from a 5C model perspective, not be attentive to promotional messages intended to build their confidence in vaccines, lessen their complacency, encourage them to greater calculation, or engender a sense of collective responsibility.

The attention of people with low involvement in a subject can be captured if messages about the subject can be linked to another matter that is involving for them. This requires identifying, for those not interested in Covid-19, themes that are involving for them and that can be meaningfully linked to containing the spread of Covid-19. Examples include framing messages about being vaccinated for Covid-19 in the context of protecting families and jobs, avoiding lockdowns, and being able to travel freely overseas [43].

A further implication concerns the intrinsic malleability of the beliefs and attitudes of people who have low involvement with a subject. Such people devote little time and effort to gathering information about the subject, evaluating that information, and forming beliefs about and attitudes towards the subject. Their search for information (calculation in the 5C model) is particularly limited. This means their beliefs and attitudes may be unstable and can change rapidly.

With respect to preventing the spread of Covid-19, this raises the possibility that, on the one hand, the distribution of misinformation about Covid-19 vaccinations through social media may provoke changes in the beliefs and attitudes of people with low involvement in Covid-19 that are undesirable because they undermine willingness to be vaccinated [12, 44, 47]. Such misinformation may provide a self-serving rationale for failing to seek vaccination should that require an investment of time and effort. On the other hand, people with low involvement are unlikely to strongly endorse misinformation (unless it is framed within a context they find highly involving) and so are unlikely to be provoked into engaging in non-compliant behaviours that require an investment of time and effort (such as attending protest rallies). However, in the context of vaccinating for infectious diseases like Covid-19, even inaction (failing to be vaccinated) can be damaging to the community.

The last implication we will consider concerns the intrinsic fixedness of the attitudes of people who have high involvement with a subject once their attitudes are established. In principle, these people have devoted substantial time and effort to gathering information about the subject, evaluating that information, and then forming beliefs about and, subsequently, formulating an attitude towards the subject. In other words, they have engaged in an extensive search for information. Consequently, their attitudes are stable and resistant to change. These people may well engage in motivated reasoning [10–12]. Hence, those who have high involvement with, and a favourable attitude towards, being vaccinated will resist misinformation about Covid-19. Those who have high involvement with, but an unfavourable attitude towards, being vaccinated will be more likely to embrace misinformation.

People with high involvement who are ambivalent about vaccination may be influenced by misinformation. However, these people are not uninformed. From a 5C model perspective, their ambivalence is likely to be a product of tension between the threat of infection (complacency) and fears about the safety of vaccines (confidence). Consequently, they will probably be responsive to promotional messages reminding them of the dangers posed by Covid-19 (complacency) while reassuring them about the safety of vaccines (confidence) and appealing to their sense of collective responsibility. The emergence of more dangerous variants of Covid-19 such as delta is likely to increase the receptivity of people in this quadrant to messaging and diminish their hesitancy [48]. These considerations suggest that government authorities must be careful to discriminate between audiences on social media in terms of involvement when it comes to investing resources in combating misinformation about Covid-19.

The willingness of people to adopt behaviours such as being vaccinated for infectious diseases like Covid-19 has been the subject of numerous studies [1–4, 6–8]. These studies have shown that willingness to be vaccinated does depend on people's attitudes, which in turn depend on their beliefs about the advantages and disadvantages of vaccinations. Consequently, many of these studies recommend that the adoption of preventive behaviours can be improved through promotional efforts intended to change beliefs and attitudes. Our results have two important implications for such recommendations.

First, while it is undoubtedly true that changing attitudes can change behaviour, promotional efforts intended to change beliefs and attitudes about vaccination are unlikely to meet with complete success unless most people have moderate-to-high involvement both with preventing the spread of Covid-19 and with being vaccinated. Fortunately, this is the case in New Zealand.

Second, the greater the proportion of people with low-to-mild involvement with preventing the spread of Covid-19 and with getting vaccinated, the more likely will efforts to improve the ease and convenience of getting vaccinated be effective in changing behaviour compared to promotional efforts aimed at doing so by changing people's beliefs and attitudes.

Our findings are subject to several qualifications. First, beliefs, attitudes and behaviours regarding preventing the spread of Covid-19 and getting vaccinated for Covid-19 may have changed over time with the emergence of the delta and omicron variants as the pandemic has progressed. Second, the survey sample was drawn from an internet panel, so there may be selection bias. While the nature and severity of this bias in relation to the beliefs, attitudes and involvement we investigated is unknown, it does seem reasonable to suppose, ceteris paribus, that people with low-to-mild involvement may be under-represented in the sample. If so, this would reinforce the importance of our cautions with respect to choosing actions to increase the uptake of vaccination.

Third, as the scales measuring willingness to be vaccinated for Covid-19 were self-reported, our measurements may have been affected by social desirability bias [49]. Public awareness of the widely promoted social desirability of getting vaccinated for Covid-19 creates the

possibility that self-reported behaviour regarding willingness to be vaccinated may be biased simply because it is socially desirable. Daoust et al. [49] found that this was the case with respect to self-reported behaviour such as mask wearing and social distancing in the context of Covid-19. However, the degree of bias with respect to vaccine hesitancy is likely to be small given that vaccination hesitancy is a widespread phenomenon, which implies hesitancy has a degree of social acceptability. The rate at which the acceptability of vaccine hesitancy has diminished recently, in nations like Australia, for example, also suggests that it is not regarded as an extreme, albeit starting, attitude to adopt [48].

The potential for systematic bias in self-reporting of socially desirable behaviours (or opinions) in the context of the $I_3$ Framework is unclear. On the one hand, those with low involvement in, say, eliminating Covid-19 and getting vaccinated might be expected to exhibit (downward) social desirability bias in self-reporting this behaviour because they are less motivated to actually engage in the behaviour. However, their very lack of interest means they may be quite insensitive to the opinions of others about the behaviour and so are less susceptible to social desirability bias.

On the other hand, those with high involvement in eliminating Covid-19 and getting vaccinated might be expected to exhibit (upward or downward) social desirability bias in self-reporting behaviours, depending on their attitude, because they are motivated to actually engage in the behaviour. Their strong interest may mean they are quite sensitive to the opinions of others about the behaviour (especially if expressing self-identity is an important source of involvement), so they may be more susceptible to social desirability bias. Through this argument, self-reporting of socially desirable behaviours like getting vaccinated might mean the actual association between involvement and engagement in such behaviours could be over-estimated or under-estimated. Clearly, the interaction between involvement and social desirability bias remains an important area for further investigation.

Overall, the degree of bias in self-reporting is likely to be small given that vaccination hesitancy is a widespread phenomenon, which implies it has a degree of social acceptability. Also, the deliberate, non-leading design of our questionnaire, by offering various degrees of compliance with relevant social norms, may be less likely to trigger guilt about breaching them. (Further work will clarify the impact of the bias on the $I_3$ Framework).

Fourth, the adoption of behaviours such as being vaccinated has been associated with a range of variables, including those relating to complacency, such as the perceived risk of infection, the local incidence rate of Covid-19, and feelings of stress in relation to Covid-19 (see [46] for example). We did not include these variables in our analysis, and, while the correlation between these variables and involvement is unknown, it is likely to be positive.

## Conclusions

Governments around the world are seeking to slow the spread of Covid-19 by implementing mass vaccination programmes. The success of these programmes depends on the commitment of individuals to respond and participate. Hence, understanding and predicting the motivation of individuals to participate is critical for designing measures to maximise participation and ensure the effectiveness of these measures in slowing the spread of the virus.

Kaine et al. [13] hypothesised that the propensity of individuals to change their behaviour and comply with policy measures depends on the intensity of their involvement with, and their attitude towards, the measure. This is because cognitive effort is required to form a strongly held attitude, and such effort is only invested when the matter at hand is sufficiently important to the individual. They also hypothesised that the propensity of individuals to comply with policy measures also depends on their involvement with the policy outcome the

measure addresses. An implication of these hypotheses is that individuals with similar attitudes will display varying degrees of compliance with policy measures depending on the intensity of their involvement with the policy outcome and the policy measure.

We tested these hypotheses, and their implications, with respect to public willingness to be vaccinated to prevent the spread of Covid-19 in New Zealand. Broadly speaking, the hypotheses and their implications were supported by the results. The finding that willingness to be vaccinated depends on involvement (motivation) as well as attitude has important implications for the design of policy measures intended to promote high vaccination rates. This finding also has important implications for the design of promotional programmes intended to encourage the participation of the community in mass vaccination programmes.

With respect to preventing the spread of Covid-19 in New Zealand, the results highlight the importance of distinguishing between those with a lack of interest in getting vaccinated from those who strongly oppose vaccination, and therefore deliberately choose not to be vaccinated, and tailoring incentive and enforcement strategies appropriately. The results also highlight the difficulty of communicating effectively through mass media with those who have low involvement with preventing the spread of Covid-19, and the importance of distinguishing between those with low and high involvement in considering the possible effects on vaccination beliefs, and vaccination rates, of the dissemination of misinformation about Covid-19 through social media.

## Supporting information

**S1 File. Questionnaire.**
(DOCX)

**S2 File. Data.**
(XLSX)

**S1 Appendix.**
(PDF)

**S2 Appendix.**
(PDF)

## Acknowledgments

We would sincerely like to thank the panellists throughout New Zealand who completed the questionnaire. Thanks also to our two anonymous referees for their constructive advice and suggestions.

## Author Contributions

**Conceptualization:** Geoff Kaine, Vic Wright, Suzie Greenhalgh.

**Data curation:** Geoff Kaine.

**Formal analysis:** Geoff Kaine, Vic Wright.

**Funding acquisition:** Geoff Kaine, Suzie Greenhalgh.

**Investigation:** Geoff Kaine, Vic Wright, Suzie Greenhalgh.

**Methodology:** Geoff Kaine, Vic Wright, Suzie Greenhalgh.

**Project administration:** Geoff Kaine.

**Resources:** Geoff Kaine.

**Writing – original draft:** Geoff Kaine, Vic Wright, Suzie Greenhalgh.

**Writing – review & editing:** Geoff Kaine, Vic Wright, Suzie Greenhalgh.

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
