## [Decision Letter · Decision Letter 0]

21 Dec 2021

PONE-D-21-33647Predicting willingness to be vaccinated for Covid-19: evidence from New ZealandPLOS ONE

Dear Dr. Kaine,

Thank you for submitting your manuscript to PLOS ONE. After careful consideration, we feel that it has merit but does not fully meet PLOS ONE’s publication criteria as it currently stands. Therefore, we invite you to submit a revised version of the manuscript that addresses the points raised during the review process.

Specifically, in the revised version of the paper please better describe the framework used and the conducted analysis.The writing flow in some key parts of the paper - quality of the analysis and concluding remarks - is very difficult to follow. When submitting the revised version of the paper, please consider the reviewers' comments. 

We look forward to receiving your revised manuscript.

Kind regards,

Camelia Delcea

Academic Editor

PLOS ONE

Journal Requirements:

Reviewers' comments:

Reviewer's Responses to Questions

**Comments to the Author**

1. Is the manuscript technically sound, and do the data support the conclusions?

Reviewer #1: No

Reviewer #2: Yes

Reviewer #3: Yes

2. Has the statistical analysis been performed appropriately and rigorously? 

Reviewer #1: I Don't Know

Reviewer #2: I Don't Know

Reviewer #3: Yes

3. Have the authors made all data underlying the findings in their manuscript fully available?

Reviewer #1: No

Reviewer #2: Yes

Reviewer #3: Yes

4. Is the manuscript presented in an intelligible fashion and written in standard English?

Reviewer #1: No

Reviewer #2: No

Reviewer #3: Yes

5. Review Comments to the Author

Reviewer #1: Key elements such as appropriate references, or figures and tables are missing

The quality of the language is not sufficient for review to take place

Tables and figures are not clear enough to read

The paper doesn’t conform to the journal’s Author Guidelines

The presentation of the paper is very poor

Reviewer #2: Review for PONE-D-21-33647

The study explores people’s attitudes towards the COVID-19 vaccine in New Zealand using a relatively new motivational framework, the I3. While I think the study has potential, it was a little difficult to follow. Therefore, I believe some revisions are required before publishing the paper.

Specific comments

1. I found the discussion of the I3 framework a little difficult to follow until I got to Figure 1 and the examples in the context of vaccine hesitancy. Perhaps the authors could provide the example of vaccine hesitancy together with their explanation rather than after it. Alternatively, they can provide a briefer general explanation and then give examples sooner.

2. The authors transition from discussing their framework to COVID-19 in New Zealand to the methods section. Since the paragraph explaining the study itself is on page 4, nine pages earlier, the reader may forget the purpose of the study. I suggest adding a short paragraph explaining what was done and why before turning to the methods section.

3. The paragraph on instrumentation was a little confusing. The authors say that there was a scale measuring involvement, but then that the order of the involvement, attitude, and belief scales was randomized. How were attitudes and belief measured? Then, the authors discuss how the attitudes scores were calculated but I could not find the scales used. It would be easier to follow if the authors had a separate paragraph for each measure and the traditional sections for instrumentation, sample, procedure, etc., though the methods section could be made clearer without using this format.

4. The authors provided information on the status of COVID-19 measures in New Zealand at the time of data collection, which is important. What was the vaccination status at the time? Were vaccines available at the time?

5. The authors wrote several paragraphs about self-report bias in the methods section. While this presents an interesting question, usually it belongs in the discussion section. Moving these paragraphs to the discussion will again make the manuscript clearer.

6. I did not understand how the respondents were classified into belief segments. Ward’s method implies a cluster analysis was used, is that so? Was this method used in other contexts using this scale, and were any validation measures used? This seems to be the main finding of the study and the results describe the clustering, so it should be presented in more detail.

7. I suggest adding standard deviations to the tables when relevant.

8. The belief segments section explains why the study was conducted and the analytic approach. I think these should be in the introduction and methods section, respectively, as exploring the belief segments is a part of the study, again adding the aforementioned details about the cluster analysis. Alternatively, the authors may split the results into “study 1” and “study 2” if they feel like the studies are separate. There are similar problems in other parts of the results section.

9. I suggest reporting the standardized coefficients in the regression tables to facilitate interpretability.

Minor comments

1. On page 4, I find it odd that the authors discuss the results before presenting them. I think the authors should discuss the framework and its importance, not specific results.

2. Figure 1 should be of higher quality.

Reviewer #3: Please put more specific information on the use of each statistical test/method used in the analysis in the Materials and Methods section, such as the use of Chronbach's Alpha (page 19, line 429), Chi-Square test (pages 19-20, 22, lines 436, 441, 448, 457, 496-498), Tukey's HSD test (pages 22, 23, lines 488, 495), Binary Logistic Regression (page 38, lines 746, 747, 748), Nagelkerke R Square (page 38, line 753), to give some knowledge to the audience/readers so that your research may be more reproducible.

6. PLOS authors have the option to publish the peer review history of their article (what does this mean?). If published, this will include your full peer review and any attached files.

Reviewer #1: No

Reviewer #2: No

Reviewer #3: No

---

## [Author Response · Author response to Decision Letter 0]

13 Jan 2022

Dear Editor,

Thank you for providing us with the opportunity to revise and resubmit our manuscript. I have entered a revised manuscript with and without ‘track’ changes as requested.

You raised the following you wished us to address:

• Specifically, in the revised version of the paper please better describe the framework used and the conducted analysis. The writing flow in some key parts of the paper - quality of the analysis and concluding remarks - is very difficult to follow. When submitting the revised version of the paper, please consider the reviewers' comments. 

We have revised the manuscript in accord with reviewers’ comments, focussing in particular on the methods and results sections. Our revisions to the reviewers’ comments are described in detail below.

Please do not hesitate to contact us if there is anything else you require.

Kind regards

Dr Geoff Kaine. 

Reviewer #2 comments:

1. I found the discussion of the I3 framework a little difficult to follow until I got to Figure 1 and the examples in the context of vaccine hesitancy. Perhaps the authors could provide the example of vaccine hesitancy together with their explanation rather than after it. Alternatively, they can provide a briefer general explanation and then give examples sooner. 

We had provided examples with respect to vaccine hesitancy in the Theory section where we introduce the I3 model. We have made the link between vaccine hesitancy and the I3 model clearer by referring specifically to the 5C model in that section. See revised lines 95, 100, 191, 213 and 268 (in the manuscript with track changes).

2. The authors transition from discussing their framework to COVID-19 in New Zealand to the methods section. Since the paragraph explaining the study itself is on page 4, nine pages earlier, the reader may forget the purpose of the study. I suggest adding a short paragraph explaining what was done and why before turning to the methods section.

We have added a short paragraph at the commencement of the Methods sections describing the purpose of the study as suggested (lines 308-312 in the manuscript with track changes). We have also added a description of the steps in the analysis, and the purpose of each step, to the Methods section (lines 432-463 in the manuscript with track changes).

3. The paragraph on instrumentation was a little confusing. The authors say that there was a scale measuring involvement, but then that the order of the involvement, attitude, and belief scales was randomized. How were attitudes and belief measured? Then, the authors discuss how the attitudes scores were calculated but I could not find the scales used. It would be easier to follow if the authors had a separate paragraph for each measure and the traditional sections for instrumentation, sample, procedure, etc., though the methods section could be made clearer without using this format.

We have added additional material to the Methods section to provide more clarity on measurement methods (lines 320-324, 329-349 in the manuscript with track changes). 

4. The authors provided information on the status of COVID-19 measures in New Zealand at the time of data collection, which is important. What was the vaccination status at the time? Were vaccines available at the time?

We have revised the relevant paragraph (lines 300-305 in manuscript with track changes) to be clear the study was conducted before vaccines were made available to the general public in New Zealand.

5. The authors wrote several paragraphs about self-report bias in the methods section. While this presents an interesting question, usually it belongs in the discussion section. Moving these paragraphs to the discussion will again make the manuscript clearer.

We have shifted the relevant paragraphs to the Discussion section (lines 998-1036 in the manuscript with track changes).

6. I did not understand how the respondents were classified into belief segments. Ward’s method implies a cluster analysis was used, is that so? Was this method used in other contexts using this scale, and were any validation measures used? This seems to be the main finding of the study and the results describe the clustering, so it should be presented in more detail.

We have revised the Methods section (lines 441-450 in the manuscript with track changes) and the reporting of the belief segments results to make it clear that cluster analysis was employed (lines 583, 594, 609, 621 in the manuscript with track changes). We have also added material to the Methods section to explain the steps in the analysis (lines 432-463 in the manuscript with track changes) and to clarify the purpose of the belief segment analysis was simply to condense the belief data into a smaller set of dummy variables for use in the regression analyses. This clarifies that the main finding of the study concern the regression results (and not the belief segment results).

7. I suggest adding standard deviations to the tables when relevant.

In our opinion adding standard deviations to the tables is unnecessary as we already report significance test results in all the relevant tables (which are themselves based on standard deviations), thereby saving the reader the effort of interpreting the results themselves. 

8. The belief segments section explains why the study was conducted and the analytic approach. I think these should be in the introduction and methods section, respectively, as exploring the belief segments is a part of the study, again adding the aforementioned details about the cluster analysis. Alternatively, the authors may split the results into “study 1” and “study 2” if they feel like the studies are separate. There are similar problems in other parts of the results section.

We have added material to the Methods section to explain the steps in the analysis (lines 432-463 in the manuscript with track changes) and to clarify the purpose of the belief segment analysis was to condense the belief data into a smaller set of dummy variables for use in the regression analyses. Hence splitting the results into separate studies was not appropriate. We have revised the opening paragraphs of the belief segments section accordingly (lines 574-581 in the manuscript with track changes).

9. I suggest reporting the standardized coefficients in the regression tables to facilitate interpretability.

We have revised the relevant tables to report standardised regression coefficients for OLS regressions (Tables 9-13).

Minor comments

1. On page 4, I find it odd that the authors discuss the results before presenting them. I think the authors should discuss the framework and its importance, not specific results.

We have removed the two paragraphs describing the results from the Introduction (lines 73-91 in the manuscript with track changes).

2. Figure 1 should be of higher quality.

We have revised the Figure.

Reviewer #3 comments:

Please put more specific information on the use of each statistical test/method used in the analysis in the Materials and Methods section, such as the use of Cronbach's Alpha (page 19, line 429), Chi-Square test (pages 19-20, 22, lines 436, 441, 448, 457, 496-498), Tukey's HSD test (pages 22, 23, lines 488, 495), Binary Logistic Regression (page 38, lines 746, 747, 748), Nagelkerke R Square (page 38, line 753), to give some knowledge to the audience/readers so that your research may be more reproducible.

We have indicated where we have used these tests and methods in the additional material in the Methods section (lines 432-463 in the manuscript with track changes).

---

## [Decision Letter · Decision Letter 1]

18 Feb 2022

PONE-D-21-33647R1Predicting willingness to be vaccinated for Covid-19: evidence from New ZealandPLOS ONE

Dear Dr. Kaine,

Thank you for submitting your manuscript to PLOS ONE. After careful consideration, we feel that it has merit but does not fully meet PLOS ONE’s publication criteria as it currently stands. Therefore, we invite you to submit a revised version of the manuscript that addresses the points raised during the review process. Thank you for the revised version of the paper. Please consider the reviewer comments when re-submitting the paper.

*Please note that reviewer 1 has requested citations which have limited relevance to the study topic. As such we suggest that you carefully review whether the suggested references further contribute to the literature discussion of your study. Please feel free to ignore any referenced without direct relevance.*<o:p></o:p>

We look forward to receiving your revised manuscript.

Kind regards,

Lucinda Shen 

Staff Editor 

on behalf of 

Camelia Delcea

Academic Editor

PLOS ONE

Journal Requirements:

Reviewers' comments:

Reviewer's Responses to Questions

**Comments to the Author**

1. If the authors have adequately addressed your comments raised in a previous round of review and you feel that this manuscript is now acceptable for publication, you may indicate that here to bypass the “Comments to the Author” section, enter your conflict of interest statement in the “Confidential to Editor” section, and submit your "Accept" recommendation.

Reviewer #1: All comments have been addressed

Reviewer #2: All comments have been addressed

Reviewer #3: All comments have been addressed

2. Is the manuscript technically sound, and do the data support the conclusions?

Reviewer #1: Yes

Reviewer #2: Yes

Reviewer #3: Yes

3. Has the statistical analysis been performed appropriately and rigorously? 

Reviewer #1: Yes

Reviewer #2: Yes

Reviewer #3: Yes

4. Have the authors made all data underlying the findings in their manuscript fully available?

Reviewer #1: Yes

Reviewer #2: Yes

Reviewer #3: Yes

5. Is the manuscript presented in an intelligible fashion and written in standard English?

Reviewer #1: Yes

Reviewer #2: Yes

Reviewer #3: Yes

6. Review Comments to the Author

Reviewer #1: Kindly, cite these references related to microorganism propulsion mechanism:

1. Dynamical interaction effects on soft-bodied organisms in a multi-sinusoidal passage. Eur. Phys. J. Plus, 136 (6) (2021), 1-17

2. , Locomotion of an efficient biomechanical sperm through viscoelastic medium. Biomech. Model Mechanobiol., 19 (2020), 2271-2284

3. An implicit finite difference analysis of magnetic swimmers propelling through non-Newtonian liquid in a complex wavy channel, Comput. Math. Appl., 79 (8) (2020) 2189-2202

4. Analytical and numerical study of creeping flow generated by active spermatozoa bounded within a declined passive tract. Eur. phys. j. plus, 134 (2019) 9

5. a mathematical model of the locomotion of bacteria near an inclined solid substrate: effects of different waveforms and rheological properties of couple stress slime. Can. J. Phys., 97 (2019) 537-547.

6. Magnetic microswimmers propelling through biorheological liquid bounded within an active channel. J. Magn. Magn. Mater., 486 (2019) 165283.

7. A hybrid numerical study of bacteria gliding on a shear rate-dependent slime. Physica A: Stat. Mech. Appl.. 535 (2019) 122435.

Reviewer #2: Review for PONE-D-21-33647-R1

I would like to thank the authors for their extensive work on this manuscript. It is now much clearer, and therefore I am able to provide more specific feedback. Overall, this study is interesting and valuable. I believe some clarifications are needed, and will greatly improve the manuscript.

Specific comments

1. The paper still suffers from minor organization issues. For example, the authors introduce the 5C framework but then say that the study uses the I3 framework, without explaining what it is and how they are connected until much later, or discuss the 5C model among people in quadrant 3 with unfavorable attitudes towards the measure before discussing those attitudes.

2. The way I understand Kaine et al. (2010), involvement with the issue in this context does not necessarily mean wanting to prevent the spread of COVID, but rather interest and research on the topic and involvement in decision-making related to it. One might conduct their own research and deduce that COVID is not very dangerous, an option addressed in the scales used in this study. This places them in the third quadrant (with unfavorable views), however, the description of this quadrant does not fit them. Could the authors please clarify the meaning of involvement with the issue, and if I understood correctly, adjust the quadrants’ descriptions accordingly?

3. On page 7 line 158: “because people in this quadrant are uninterested, they are unlikely to pay attention to promotional and educational messaging, so the final strategy of promoting awareness is likely to be problematic.” I do not understand why promoting awareness is likely to be problematic. Do the authors mean ineffective? Or that promoting awareness is likely to cause resistance to the measure? This can be made clearer.

4. On page 8 line 171: it seems to me that people with high involvement with the policy outcome will not be complacent with respect to the risk. They are familiar for example with COVID and its risks and are concerned about it, but are not familiar with the vaccine as the intervention. Perhaps the authors should explain why they think people in the different quadrants have these levels of the C5.

5. On page 9 line 188: “Consequently, a strategy for promoting compliance among individuals in this quadrant with a favourable attitude might focus on self-regulation, using mechanisms such as voluntary codes of conduct.” I am not sure I understand what voluntary codes of conduct mean in this case, and why they are necessary if the people in question are already likely to comply.

6. In quadrants 3 and 4, the authors assume that collective responsibility involves getting vaccinated. This may not be the case; some of those opposing the vaccine believe it is better for people to not get vaccinated (e.g., they believe the vaccines are dangerous to young people, again an option mentioned in the current survey). If the authors argue that collective responsibility is only linked to adopting the measure rather than other altruistic intentions (even if misguided), they should clarify this point.

7. I believe that the discussion of enforcing vaccines should be done more carefully given the current debate on the issue. The other alternatives for increasing compliance among Q3 people with unfavorable attitudes towards COVID vaccines are dismissed as ineffective, but not enforcement. This topic is well-covered in the discussion, so I think the introduction should also present the compliance issues with enforcement.

8. It is unclear to me how someone can have favorable attitudes towards the COVID-19 vaccine without being involved with wanting to prevent the disease. Perhaps the authors could provide an example?

9. I thank the authors for providing the information about vaccinations in New Zealand.

10. The authors introduce the five components of involvement in the methods section, but this was not mentioned before. If the authors are not interested in presenting these components in detail in the introduction, they can briefly describe each component in the methods.

11. Based on the description of the measures, it seems like there was only one measure of involvement. The supplement material has two involvement scales, which makes it clear how the quadrants were determined. In order to help the reader who may not read the supplement material, I suggest clarifying that both types of involvement were measured.

12. In the first stage of analysis, there seems to be no separation between favorable and unfavorable attitudes towards the vaccine in the relevant quadrants. Was this planned or due to the small proportion of unfavorable attitudes in the sample?

13. I suggest that the authors use item means in their tables instead of the scores on the individual items. This will help reduce the number of tables and make them clearer.

14. On page 28: since the authors explain the clustering technique in the methods section, they do not have to mention it again in the results section.

15. I thank the authors for including the standardized coefficients in their tables reporting the regression results. Note that when reporting standardized coefficients, the intercept is always zero. If the authors want to include the intercept they should have separate columns for standardized and unstandardized parameters.

16. If the benchmark group is skeptics, then the results’ description should reflect that. For example, respondents with ambivalent beliefs about COVID had less favorable attitudes in comparison to the skeptics, not all of the other respondents.

Reviewer #3: (No Response)

7. PLOS authors have the option to publish the peer review history of their article (what does this mean?). If published, this will include your full peer review and any attached files.

Reviewer #1: **Yes: **Dr. Zeeshan Asghar

Reviewer #2: No

Reviewer #3: No

---

## [Author Response · Author response to Decision Letter 1]

1 Mar 2022

Reviewer #2 comments:

1. The paper still suffers from minor organization issues. For example, the authors introduce the 5C framework but then say that the study uses the I3 framework, without explaining what it is and how they are connected until much later, or discuss the 5C model among people in quadrant 3 with unfavorable attitudes towards the measure before discussing those attitudes.

We have replaced the discussion of the 5C model in the Introduction with a short statement referring to vaccine hesitancy generally. We introduce the components of vaccine hesitancy in the description of the I3 Framework (quadrant 1 in particular) to assist the reader to better understand the implications of Framework. See revised lines 46-61, 124-125 and 143-159 (in the manuscript with track changes). We hope the restructure improves the readability of the manuscript.

Regarding attitudes towards being vaccinated among people in quadrant 3. The discussion here is intended to illustrative and we are suggesting some hypotheses about these people regarding the components of vaccine hesitance, simply on the basis of their involvement and whether their attitude is favourable or unfavourable. Hopefully we have made the illustrative nature of this discussion clearer by amending lines 214-215 (in the manuscript with track changes).

2. The way I understand Kaine et al. (2010), involvement with the issue in this context does not necessarily mean wanting to prevent the spread of COVID, but rather interest and research on the topic and involvement in decision-making related to it. One might conduct their own research and deduce that COVID is not very dangerous, an option addressed in the scales used in this study. This places them in the third quadrant (with unfavorable views), however, the description of this quadrant does not fit them. Could the authors please clarify the meaning of involvement with the issue, and if I understood correctly, adjust the quadrants’ descriptions accordingly?

We believe the reviewer is confusing involvement with Covid-19 per se and involvement with the policy outcome of eliminating Covid-19 from New Zealand. We have revised the definition of involvement with the policy outcome to clarify we are considering involvement with the outcome of elimination (lines 106-109 in the manuscript with track changes). Consequently, the quadrant descriptions do not require changing.

3. On page 7 line 158: “because people in this quadrant are uninterested, they are unlikely to pay attention to promotional and educational messaging, so the final strategy of promoting awareness is likely to be problematic.” I do not understand why promoting awareness is likely to be problematic. Do the authors mean ineffective? Or that promoting awareness is likely to cause resistance to the measure? This can be made clearer.

We have changed ‘problematic’ to ‘ineffective’ (line 177 in the manuscript with track changes). 

4. On page 8 line 171: it seems to me that people with high involvement with the policy outcome will not be complacent with respect to the risk. They are familiar for example with COVID and its risks and are concerned about it, but are not familiar with the vaccine as the intervention. Perhaps the authors should explain why they think people in the different quadrants have these levels of the C5

We have revised the relevant paragraph (lines 191-192 in manuscript with track changes) to include reasons why people with high involvement in the policy outcome may be complacent with respect to risk of infection.

5. On page 9 line 188: “Consequently, a strategy for promoting compliance among individuals in this quadrant with a favourable attitude might focus on self-regulation, using mechanisms such as voluntary codes of conduct.” I am not sure I understand what voluntary codes of conduct mean in this case, and why they are necessary if the people in question are already likely to comply.

Codes of conduct may be an appropriate mechanism in other contexts (e.g. occupational health and safety) but are not relevant in this context as the reviewer points out. We thought the simplest solution was to delete mention of this mechanism from the paragraph (line 209 in the manuscript with track changes).

6. In quadrants 3 and 4, the authors assume that collective responsibility involves getting vaccinated. This may not be the case; some of those opposing the vaccine believe it is better for people to not get vaccinated (e.g., they believe the vaccines are dangerous to young people, again an option mentioned in the current survey). If the authors argue that collective responsibility is only linked to adopting the measure rather than other altruistic intentions (even if misguided), they should clarify this point.

In models of vaccine hesitancy such as the 5C model, ‘collective responsibility’ refers specifically to ‘a willingness to protect others by getting vaccinated’ (see line 155 in the manuscript with track changes). We have added this qualifier at lines 221 and 282 in the manuscript with track changes).

7. I believe that the discussion of enforcing vaccines should be done more carefully given the current debate on the issue. The other alternatives for increasing compliance among Q3 people with unfavorable attitudes towards COVID vaccines are dismissed as ineffective, but not enforcement. This topic is well-covered in the discussion, so I think the introduction should also present the compliance issues with enforcement.

In the theory section we describe a number of alternatives for increasing compliance among Q3 with unfavourable attitudes with enforcement mentioned as one alternative. We did not conclude the alternatives to enforcement would be ineffective., though we did note they might be unsuccessful if people engage in motivated reasoning. To avoid appearing to advocate enforcement too forcefully we have qualified the suggestion of enforcement in relation to Q3 (and Q4) with unfavourable attitudes by noting the imposition of blanket penalties for non-compliance may run the risk of incidentally alienating people with low involvement (see lines 242-244 and 269-271 in the manuscript with track changes). 

We did note that if the causes of non-compliance relate to unpredictable variations in the environment, or to unforeseeable technical problems, then enforcement and general deterrence may be ineffective, and the provision of technical assistance would be more appropriate; one could imagine this might translate into alternative healthcare arrangements for example (see lines 246-249 in the manuscript with track changes).

8. It is unclear to me how someone can have favorable attitudes towards the COVID-19 vaccine without being involved with wanting to prevent the disease. Perhaps the authors could provide an example?

We presume the Reviewer is referring to the possibility of a person in quadrant 4 with a favourable attitude towards vaccination but low involvement with eliminating Covid-19. The key here is that having a favourable attitude does not necessarily mean one is highly motivated. One may have a favourable attitude towards vaccination, that does not mean one necessarily cares strongly about ‘eliminating Covid-19 from New Zealand’. For example, a person who views vaccination purely in terms of personal protection may favour vaccination but be disinterested in the outcome of eliminating Covid-19. Presumably such a person will have a limited sense of collective responsibility in the 5C sense. We have amended the relevant paragraph to include this example (see lines 277-278 in the manuscript with track changes) and thank the reviewer for bringing this opportunity to clarify to our attention.

9. I thank the authors for providing the information about vaccinations in New Zealand.

10. The authors introduce the five components of involvement in the methods section, but this was not mentioned before. If the authors are not interested in presenting these components in detail in the introduction, they can briefly describe each component in the methods.

We have added a brief definition of the five components of involvement to the Methods section (see lines 327-332 in the manuscript with track changes). 

11. Based on the description of the measures, it seems like there was only one measure of involvement. The supplement material has two involvement scales, which makes it clear how the quadrants were determined. In order to help the reader who may not read the supplement material, I suggest clarifying that both types of involvement were measured.

We have revised the relevant paragraph to be clear both types of involvement were measured (see lines 323-324 in the manuscript with track changes). 

12. In the first stage of analysis, there seems to be no separation between favorable and unfavorable attitudes towards the vaccine in the relevant quadrants. Was this planned or due to the small proportion of unfavorable attitudes in the sample?

The proportion of respondents with favourable and unfavourable attitudes towards vaccination in each quadrant is reported in Table 3. As the reviewer observes, fundamentally, the proportion of unfavourable responses in the sample was too small to generate significant results. Furthermore, the data used to generate the results reported in Tables 5 and 6 was restricted to respondents who indicated that they definitely would, probably would, or might get vaccinated.

13. I suggest that the authors use item means in their tables instead of the scores on the individual items. This will help reduce the number of tables and make them clearer.

We believe the reviewer is referring to Tables 8 and 9 where we present results showing differences across the quadrants with respect to items from the survey that we believe related to the components of the 5C model. We prefer not to summarise these tables as suggested because the variables were not intended to be scales for measuring the 5C components and are not the standard scale items used in 5C model (these qualifications are made at lines 520-522 in the manuscript with track changes). By presenting the mean scores for each variable (rather than aggregating them into scales) the reader is in a position to judge for themselves whether the variables we report reflect the 5C components and judge the results accordingly.

14. On page 28: since the authors explain the clustering technique in the methods section, they do not have to mention it again in the results section.

We have revised the relevant paragraphs in the Methods and Results to remove the repetition (see lines 427-429 and lines 563-566 in the manuscript with track changes). 

15. I thank the authors for including the standardized coefficients in their tables reporting the regression results. Note that when reporting standardized coefficients, the intercept is always zero. If the authors want to include the intercept, they should have separate columns for standardized and unstandardized parameters.

We have removed reporting of intercepts in the relevant tables (Tables 9-13 in the manuscript with track changes). 

16. If the benchmark group is skeptics, then the results’ description should reflect that. For example, respondents with ambivalent beliefs about COVID had less favorable attitudes in comparison to the skeptics, not all of the other respondents.

We have revised the relevant paragraph Results to state beliefs about Covid-19 itself did not strongly influence attitude toward eliminating Covid-19 in New Zealand over and above beliefs about elimination (see lines 736-739 in the manuscript with track changes).

---

## [Decision Letter · Decision Letter 2]

22 Mar 2022

Predicting willingness to be vaccinated for Covid-19: evidence from New Zealand

PONE-D-21-33647R2

Dear Dr. Kaine,

We’re pleased to inform you that your manuscript has been judged scientifically suitable for publication and will be formally accepted for publication once it meets all outstanding technical requirements.

Kind regards,

Sonia Brito-Costa, Ph.D.

Academic Editor

PLOS ONE

Additional Editor Comments (optional):

Reviewers' comments:

Reviewer's Responses to Questions

**Comments to the Author**

1. If the authors have adequately addressed your comments raised in a previous round of review and you feel that this manuscript is now acceptable for publication, you may indicate that here to bypass the “Comments to the Author” section, enter your conflict of interest statement in the “Confidential to Editor” section, and submit your "Accept" recommendation.

Reviewer #1: (No Response)

Reviewer #2: All comments have been addressed

2. Is the manuscript technically sound, and do the data support the conclusions?

Reviewer #1: No

Reviewer #2: Yes

3. Has the statistical analysis been performed appropriately and rigorously? 

Reviewer #1: No

Reviewer #2: Yes

4. Have the authors made all data underlying the findings in their manuscript fully available?

Reviewer #1: No

Reviewer #2: Yes

5. Is the manuscript presented in an intelligible fashion and written in standard English?

Reviewer #1: No

Reviewer #2: (No Response)

6. Review Comments to the Author

Reviewer #1: (No Response)

Reviewer #2: I thank the authors for addressing all of my comments. I believe the manuscript is now ready for publication, with the exception of a minor correction I believe to be a typo: In line 220, I think people in Q3 (and Q4) who have unfavorable attitudes should be sensitive to constraints on being vaccinated, which is one of the reasons they have unfavorable attitudes. The authors also mention that they are sensitive to the policy’s costs in the next paragraph, strengthening this point. If I misunderstood, the authors can clarify what they mean by insensitivity to costs among those groups. Otherwise, I wish the authors good luck in their future endeavors.

7. PLOS authors have the option to publish the peer review history of their article (what does this mean?). If published, this will include your full peer review and any attached files.

Reviewer #1: No

Reviewer #2: No

---

## [Editor Report · Acceptance letter]

24 Mar 2022

PONE-D-21-33647R2 

Predicting willingness to be vaccinated for Covid-19: evidence from New Zealand 

Dear Dr. Kaine:

I'm pleased to inform you that your manuscript has been deemed suitable for publication in PLOS ONE. Congratulations! Your manuscript is now with our production department. 

Kind regards, 

on behalf of

Dr. Sonia Brito-Costa 

Academic Editor

PLOS ONE